# Meta-analysis of multidecadal biodiversity trends in Europe

Francesca Pilotto [ID] et al.[#]

Local biodiversity trends over time are likely to be decoupled from global trends, as local processes may compensate or counteract global change. We analyze 161 long-term biological time series (15–91 years) collected across Europe, using a comprehensive dataset comprising ~6,200 marine, freshwater and terrestrial taxa. We test whether (i) local long-term biodiversity trends are consistent among biogeoregions, realms and taxonomic groups, and (ii) changes in biodiversity correlate with regional climate and local conditions. Our results reveal that local trends of abundance, richness and diversity differ among biogeoregions, realms and taxonomic groups, demonstrating that biodiversity changes at local scale are often complex and cannot be easily generalized. However, we find increases in richness and abundance with increasing temperature and naturalness as well as a clear spatial pattern in changes in community composition (i.e. temporal taxonomic turnover) in most biogeoregions of Northern and Eastern Europe.

[#]A list of authors and their affiliations appears at the end of the paper.

The current biodiversity crisis, manifested in a global decline of species, affects many taxonomic groups and biotic realms[1–4]. These changes may be less evident at specific locations, since local factors, such as small-scale colonization and species turnover may compensate or even counteract trends occurring at larger spatial scales. Heterogeneity in patterns of change in biodiversity observed at a local scale[5,6] has been described in several studies. For example, strong declines in local biomass and distribution have been reported for terrestrial insects[7–10] and birds[11–13], but reports of local increases in biodiversity are also widespread and span multiple taxonomic groups[5] including freshwater invertebrates[14,15], fishes[16], birds[12,13] and plants[17–19].

Ecosystem functions and their benefits to people at local to global scales ultimately depend on the taxonomic and functional diversity of local communities[5,20]. The same relationship applies to conservation measures: while they need to be harmonized at larger scales, most measures need to be tailored to local conditions. Therefore, it is crucial to understand the variation in biodiversity trends between localities and how and why these may vary across biota, regions and local conditions[21].

Analyses of the trends in local biodiversity over large spatial scales and multiple taxonomic groups are needed to fully understand the patterns of local biodiversity change and the discrepancies between local and global biodiversity trends. Unfortunately such syntheses are rare. Current evidence from multi-taxa biodiversity trend comparisons is limited and equivocal, and direct comparison across studies is hampered by substantial differences in the temporal and spatial coverage and resolution of the data. Dornelas et al.[6] studied 100 time series of terrestrial, freshwater and marine taxonomic groups around the world and found no systematic temporal changes in α-diversity of local communities. However, the authors detected significant increases in β-diversity and increasing trends in species richness for terrestrial plants in the temperate region[6]. The most comprehensive study to date to our knowledge[22] showed that species turnover, i.e., a measure of temporal community variability, is stronger in marine than freshwater and terrestrial assemblages and that this is often decoupled from changes in species richness. However, these results were based on a relatively coarse spatial

and temporal resolution, and mostly short time series. Other studies focused on a smaller number of sites or more restricted biotic scope. A study of 22 sites in different realms from Central Europe detected stronger effects of temperature on population trends in terrestrial than aquatic habitats (i.e. populations of terrestrial "species with warmer temperature preferences increased more than [terrestrial] species with colder temperature preferences" (page 3), but found no such relationship for aquatic taxa)[23]. Finally, Gibson-Reinemer et al.[24] found that the increasing species turnover in mountain communities was stronger in ectotherm communities and in tropical compared to temperate regions. These results point towards region, biotic, and realm specific patterns in local biodiversity trends, but a comprehensive overview directly comparing trends among complementary time series has been lacking so far.

The present study analyzes trends in various taxonomic groups measured at specific locations in nine different biogeoregions across the European continent. We ask: (1) are long-term trends in biodiversity detectable at individual localities across Europe? (2) If so, to what extent are such trends attributable to changes in climate at a regional scale and/or changes in local conditions? (3) Do observed trends in biodiversity vary predictably among biogeoregions, realms and taxonomic groups and, thus, inform our understanding and prediction of larger scale patterns? Due to the ongoing global change we expect to observe: (1) long-term reductions in biodiversity indicated by declining species richness and abundances (2) increasing variability in community composition, indicated by higher temporal species turnover and (3) differential responses across taxonomic groups and biogeoregions, reflecting differences in the extent of and vulnerability to climate change (i.e. Southern Europe should be more negatively affected than Northern Europe[25]).

We compile 161 long-term (minimum 15 years) biomonitoring time series from 115 sites, mostly belonging to the International Long-Term Ecological Research network (ILTER[26]), in 21 European countries (Fig. 1), covering nine biogeoregions, three realms and eight taxonomic groups (Fig. 1, Supplementary Table 1, Supplementary Data). For each time series, we compute total abundance of the community, taxonomic richness, diversity and temporal turnover and estimated their monotonic trends over the study period.

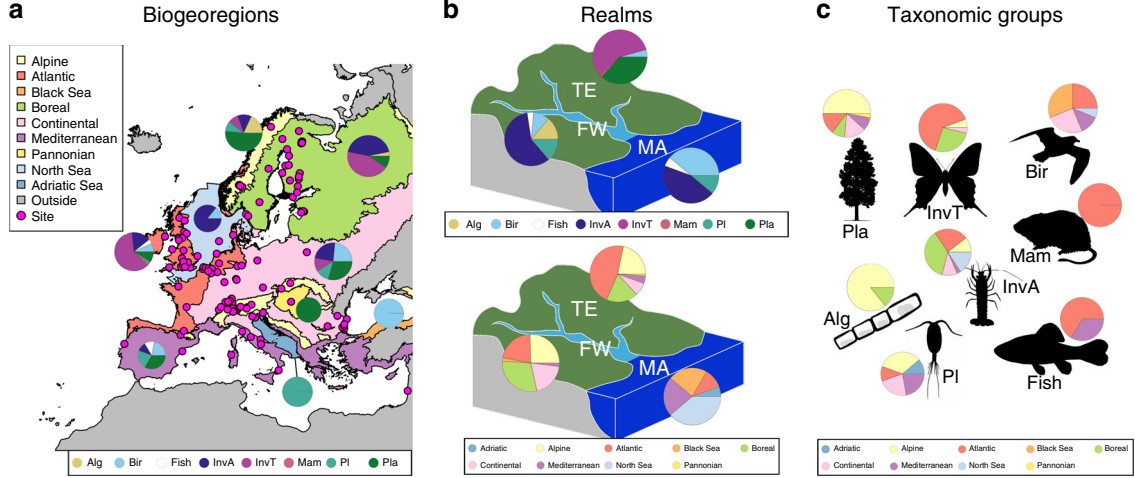

**Fig. 1 Distribution of the time series across biogeoregions, realms and taxonomic groups. a** Relative distribution of studied taxonomic groups across biogeoregions (magenta dots: study sites). Note that the most south-eastern site (in Israel) belongs to the Mediterranean region. **b** Relative distribution of studied taxonomic groups and biogeoregions across realms. **c** Relative distribution of studied biogeoregions across taxonomic groups. FW freshwater, MA marine and transitional zone, TE terrestrial, Alg benthic algae, Bir birds, InvA aquatic invertebrates, InvT terrestrial invertebrates, Mam mammals, Pl plankton, Pla terrestrial plants. The pie charts show the proportion of taxonomic groups for each biogeoregion and realm, and the proportion of biogeoregions for each realm and taxonomic group. The shapefiles of the biogeographical regions and marine subregions were obtained from EEA[74]. Drawings of taxonomic groups are from phylopic.org. Source data are provided as a Source Data file.

**Table 1 Biodiversity trends.**

|  |  | Abundance | Richness | Diversity | Turnover |
|---|---|---|---|---|---|
| *Overall trend* | z | −1.4 | 2.95 | 2 | 3.59 |
|  | d.f. | 152 | 160 | 160 | 160 |
|  | p | 0.162 | 0.003 | 0.045 | 0.003 |
| *Biogeoregion* |  |  |  |  |  |
| Wald-type test of model coefficients | QM | 24.397 | 65.83 | 21.23 | 38.6 |
|  | d.f. | 8 | 9 | 9 | 9 |
|  | p | 0.002 | <0.001 | 0.012 | <0.001 |
| Atlantic | z | −2.73 |  |  |  |
|  | p | 0.006 |  |  |  |
| North Sea | z | 2.73 | 5.77 | 2.55 | −2.42 |
|  | p | 0.006 | <0.001 | 0.011 | 0.015 |
| Black Sea | z |  | 3.41 | 2.67 | −2.73 |
|  | p |  | <0.001 | 0.008 | 0.006 |
| Boreal | z |  | 4.35 | 2.09 | 2.05 |
|  | p |  | <0.001 | 0.037 | 0.04 |
| Alpine | z |  |  |  | 3.79 |
|  | p |  |  |  | <0.001 |
| Continental | z |  |  |  | 2.49 |
|  | p |  |  |  | 0.01 |
| *Realm* |  |  |  |  |  |
| Wald-type test of model coefficients | QM | 4.17 | 22.75 | 12.58 | 13.15 |
|  | d.f. | 3 | 3 | 3 | 3 |
|  | p | 0.243 | <0.001 | 0.006 | 0.004 |
| Freshwater | z |  | 2.73 |  |  |
|  | p |  | 0.006 |  |  |
| Marine | z |  | 3.9 | 3.37 |  |
|  | p |  | <0.001 | <0.001 |  |
| Terrestrial | z |  |  |  | 3.25 |
|  | p |  |  |  | 0.001 |
| *Taxonomic group* |  |  |  |  |  |
| Wald-type test of model coefficients | QM |  | 24.67 | 57.09 | 20.06 |
|  | d.f. |  | 8 | 8 | 8 |
|  | p |  | 0.002 | <0.001 | 0.01 |
| Terrestrial invertebrates | z | −2.95 |  |  |  |
|  | p | 0.003 |  |  |  |
| Birds | z |  | 4.11 | 4.78 |  |
|  | p |  | <0.001 | <0.001 |  |
| Aquatic invertebrates | z |  | 2.42 | 2.26 |  |
|  | p |  | 0.015 | 0.024 |  |
| Benthic algae | z |  |  | −5.33 |  |
|  | p |  |  | <0.001 |  |
| Plants | z |  |  |  | 3.7 |
|  | p |  |  |  | <0.001 |

Biodiversity trends for the whole dataset (overall trends) and within the different biogeoregions, realms and taxonomic groups, as resulting from meta-analysis mixed models. Note that only significant results ($p \leq 0.05$) are reported for the biogeoregion, realm and taxonomic group-specific analysis.

We then apply a meta-analytic approach to identify the patterns among biogeoregions, realms and taxonomic groups. Our time series have an unbalanced distribution across biogeoregions, realms and taxonomic groups, which is a common issue in macroecological studies[6,22,23]. We account for the unbalanced design by testing the robustness of our results using a sensitivity analysis (Supplementary notes) and when interpreting the results. Our results show that trends in common biodiversity metrics differ among biogeoregions, realms, and taxonomic groups. Particularly, we find stronger changes in community composition in Northern and Eastern Europe and increases in richness and abundance with increasing temperature and naturalness.

## Results

**Overall trends**. We observed overall increasing trends in taxonomic richness, diversity and turnover, across all time series, while there was no significant trend in abundance (Table 1).

**Biodiversity trends**. The trends of all biodiversity metrics differed among biogeoregions (Table 1). Abundances declined in the Atlantic region and increased in the North Sea over time. Species richness and diversity increased in the Black Sea, Boreal region and North Sea. Species turnover increased over time in time series from the Alpine, Boreal and Continental regions, and decreased in time series from the Black Sea and North Sea (Table 1, Fig. 2).

We did not detect any clear trends in abundance in any realm, while trends in taxon richness, diversity and turnover varied across realms. We recorded increasing taxon richness in the freshwater realm, increasing taxon richness and diversity in the marine realm, and increasing turnover in the terrestrial realm (Table 1; Fig. 3).

We found a significant decline in the abundance of terrestrial invertebrates. Species richness, diversity and turnover trends differed among taxonomic groups (Table 1). Species richness and diversity increased in birds and aquatic invertebrates. This contrasted with the decreasing diversity in benthic algae (note that algae data were only available for sites within a single river catchment). Turnover trends significantly increased for plants. Other taxa did not show a trend in the respective biodiversity metrics (Table 1, Fig. 4).

**Influence of climatic trends and site characteristics**. The most important predictors (i.e., highest absolute values of z-scores) of abundance and richness trends were site naturalness (i.e., a measure of local anthropogenic pressure), temperature trend (i.e., S-statistics of air or water temperature trend, according to the realm) and their interaction (Table 2). Increases in temperature and site naturalness were correlated with increased abundance and richness. The negative interaction between site naturalness and temperature trend indicates stronger increases in abundance and richness with increasing temperature at sites with lower naturalness than at sites with high naturalness (Supplementary Fig. 1). The most important predictor of diversity trends was longitude (positive correlation). Species turnover trends were mostly affected by elevation (positive correlation) and by the positive interaction between site naturalness and temperature trend, indicating stronger increases in turnover with increasing temperature at sites with higher naturalness than at sites with low naturalness (Table 2, Supplementary Fig. 1). Study length had much less influence on the trends, indicating that the differences in study intervals among time series did not bias our results (Table 2).

## Discussion

Our results demonstrate considerable heterogeneity in the extent and direction of change in biodiversity metrics in recent decades between biogeoregions, realms and taxonomic groups in Europe. We identified increases in taxon richness in Northern and Eastern Europe. Records in these regions are primarily represented by terrestrial and aquatic invertebrate datasets. These observed patterns are in line with recent modelled predictions of future responses to global changes[25] and are likely due to climate-change induced poleward range shifts across taxa[27–30]. We also detected declines in species abundances for terrestrial invertebrates and in the Atlantic biogeoregion (from which most of the

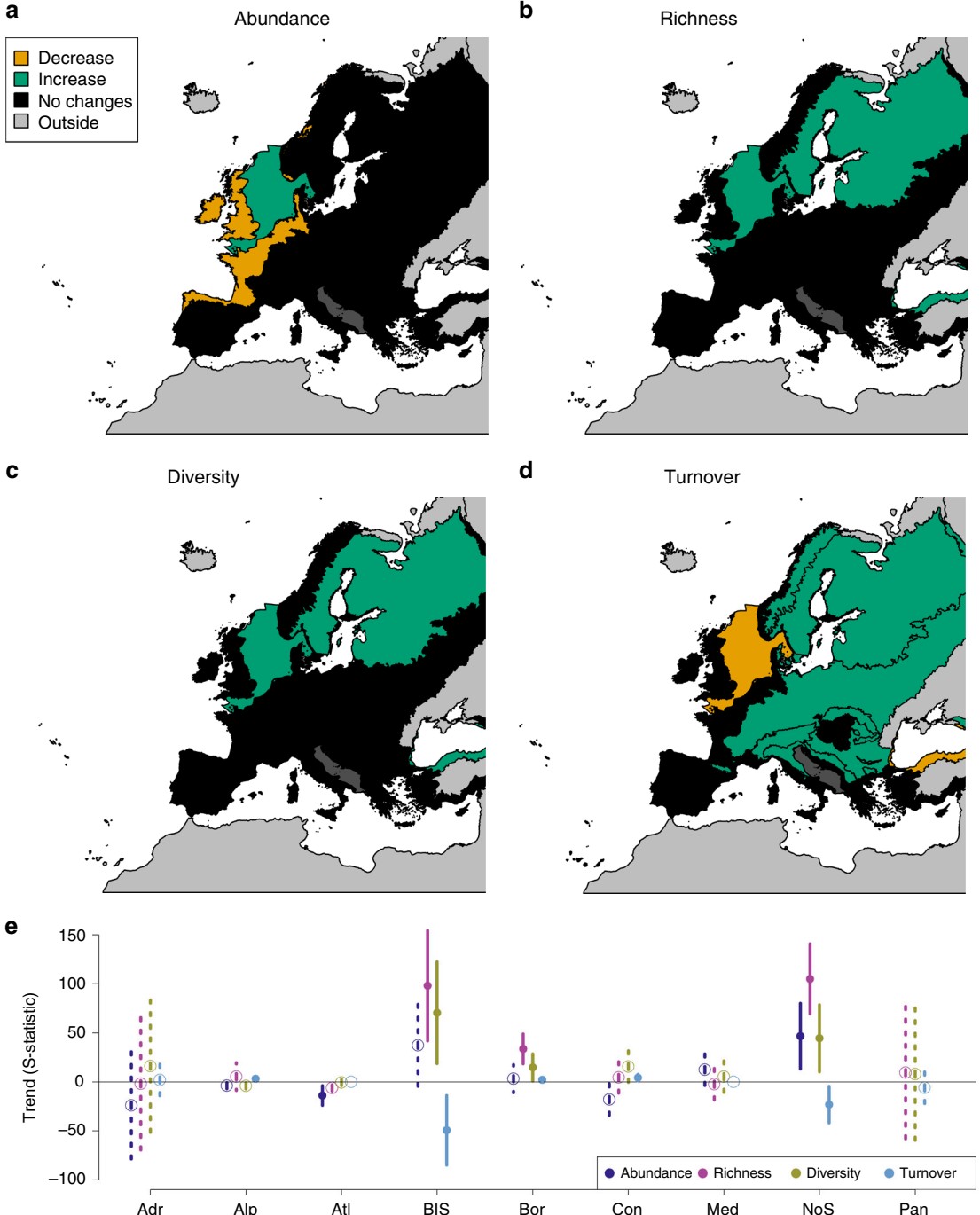

**Fig. 2 Biodiversity trends in the different biogeoregions.** The results of meta-analysis mixed models are shown for the four studied biodiversity metrics: abundance (**a**), richness (**b**), diversity (**c**) and turnover (**d**). Green: significant increasing trends ($p \leq 0.05$); orange: significantly declining trends ($p \leq 0.05$); black (dark grey for Adriatic Sea): no significant trends ($p > 0.05$). For biogeoregion identity see Fig. 1. **e** Values of S-statistics (model estimated mean, error bar: $+/-$ C.I.). Adr: Adriatic ($n = 1$ time series), Alp: Alpine ($n = 33$ time series), Atl Atlantic ($n = 56$ time series), BlS Black Sea ($n = 5$ time series), Bor Boreal ($n = 32$ time series), Con Continental ($n = 17$ time series), Med Mediterranean ($n = 9$ time series), NoS North Sea ($n = 7$ time series), and Pan Pannonian ($n = 1$ time series). Dark blue: abundance, pink: richness, yellow: diversity, light blue: turnover. Solid line and dot: $p \leq 0.05$; dashed line and open circle: $p > 0.05$. Source data are provided as a Source Data file.

terrestrial invertebrate time series in our dataset are derived). Our results, based on data with larger spatial, temporal and taxonomic coverage, and finer temporal resolution, corroborate recent reports of worldwide declines of local terrestrial insect communities[31–33].

Several studied biogeoregions, realms and biotic groups showed no significant trends in biodiversity metrics. Other

studies on local changes in biodiversity also detected no overall changes[5,22,34] in apparent contradiction to the documented global-scale biodiversity loss (e.g. IPBES[4]). However, the extinction of rare species, by definition, is often restricted to the very few local places where these species occur and may thus go undetected in large extent quantitative studies[35]. Furthermore, the loss of specialist taxa could be compensated locally by the

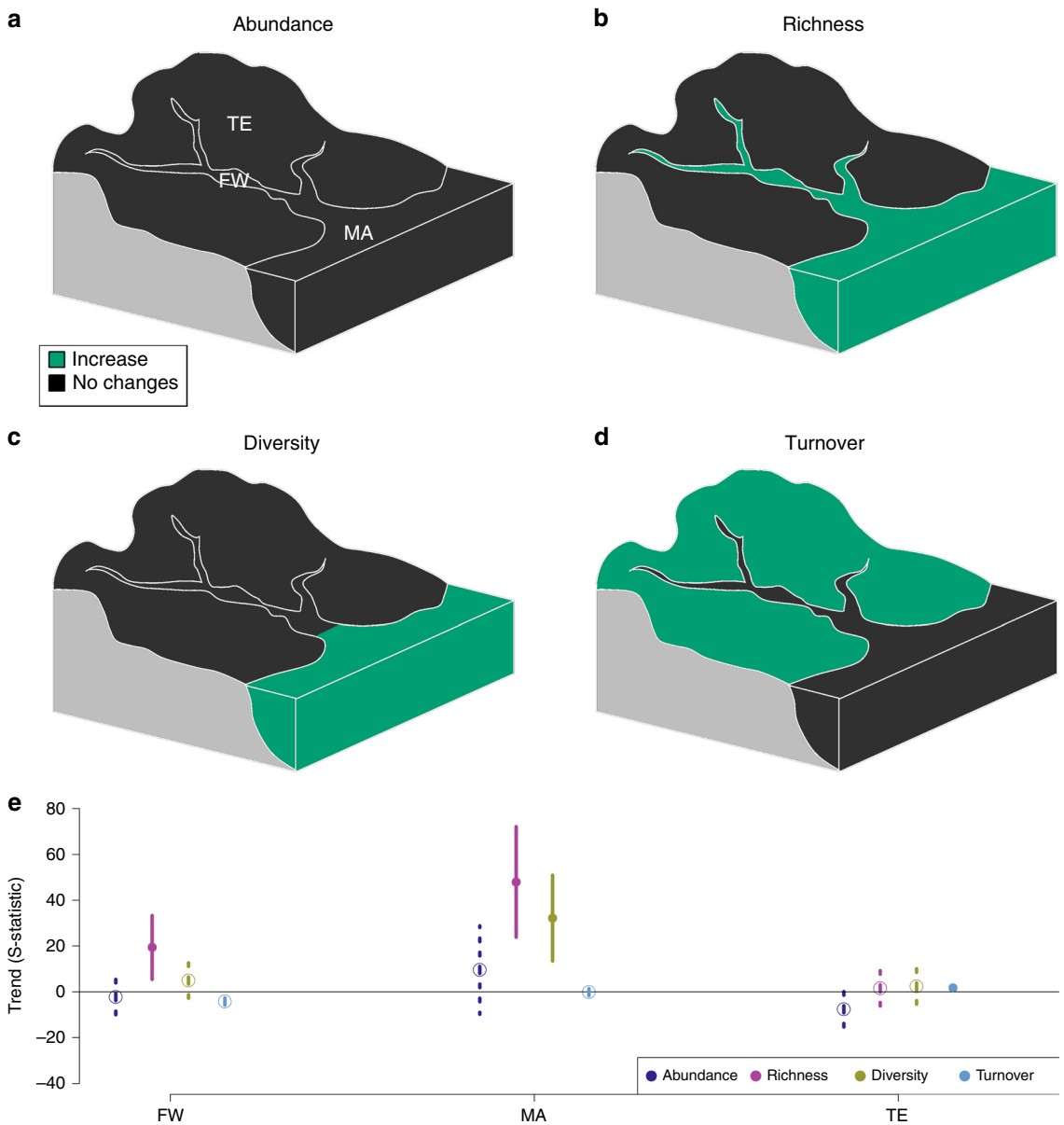

**Fig. 3 Biodiversity trends in the three realms.** The results of meta-analysis mixed models are shown for the four studied biodiversity metrics: abundance (**a**), richness (**b**), diversity (**c**) and turnover (**d**). Green: significant increasing trends ($p \leq 0.05$); black: no significant trends ($p > 0.05$). **e** Values of S-statistics (model estimated mean, error bar: $+/-$C.I.). Dark blue: abundance, pink: richness, yellow: diversity, light blue: turnover. Solid line and dot: $p \leq 0.05$; dashed line and open circle: $p > 0.05$. FW freshwater ($n = 51$ time series); MA marine and transitional zones ($n = 18$ time series); TE terrestrial ($n = 92$ time series). Source data are provided as a Source Data file.

colonization of more tolerant and generalist species[36] and/or invasive taxa, resulting in biotic homogenization despite no change in overall species richness[36,37]. Such patterns were, for example, observed in vascular plants at coarse scales in Europe due to the extinction of rare native species and spread of alien species[38]. Thus, temporal changes in taxonomic composition, i.e., turnover, are likely to be more sensitive than simply taxonomic richness (i.e., alpha-diversity) and abundance in responses to global change[39,40]. Accordingly, we show that temporal changes in taxon turnover are more pertinent across biogeoregions than the other three studied biodiversity metrics. The observed increases in temporal taxon turnover are spatially structured across Europe, involving mostly biogeoregions of Northern and Eastern Europe. Such changes in community composition can reflect non-equilibrium dynamics (including time-lags and transient phenomena[41]) with, for example, climate change[42],

pollution or the introduction and spread of alien species[43]. Moreover, over the past 30–40 years (i.e. the period covered by most time series in our dataset) positive impacts of environmental regulation, e.g., reductions in atmospheric emissions and hence acid rain, could also be important drivers of biotic change (see e.g., Monteith et al.[44]). In a recent analysis of UK vegetation data, an overall increase in vegetation species richness was linked to recovery from acidification[45]. Accordingly, increases in species turnover could also reflect a process of biological recovery from past disturbances. Another possible interacting factor is the change in land use, which follows different temporal trajectories in different European regions, and thus could concur in explaining regional differences in biodiversity trends[46]. Future research should clarify to what extent the increasing taxon turnover is led by the spread of generalist and invasive species compared to declines in rare species, and whether the observed

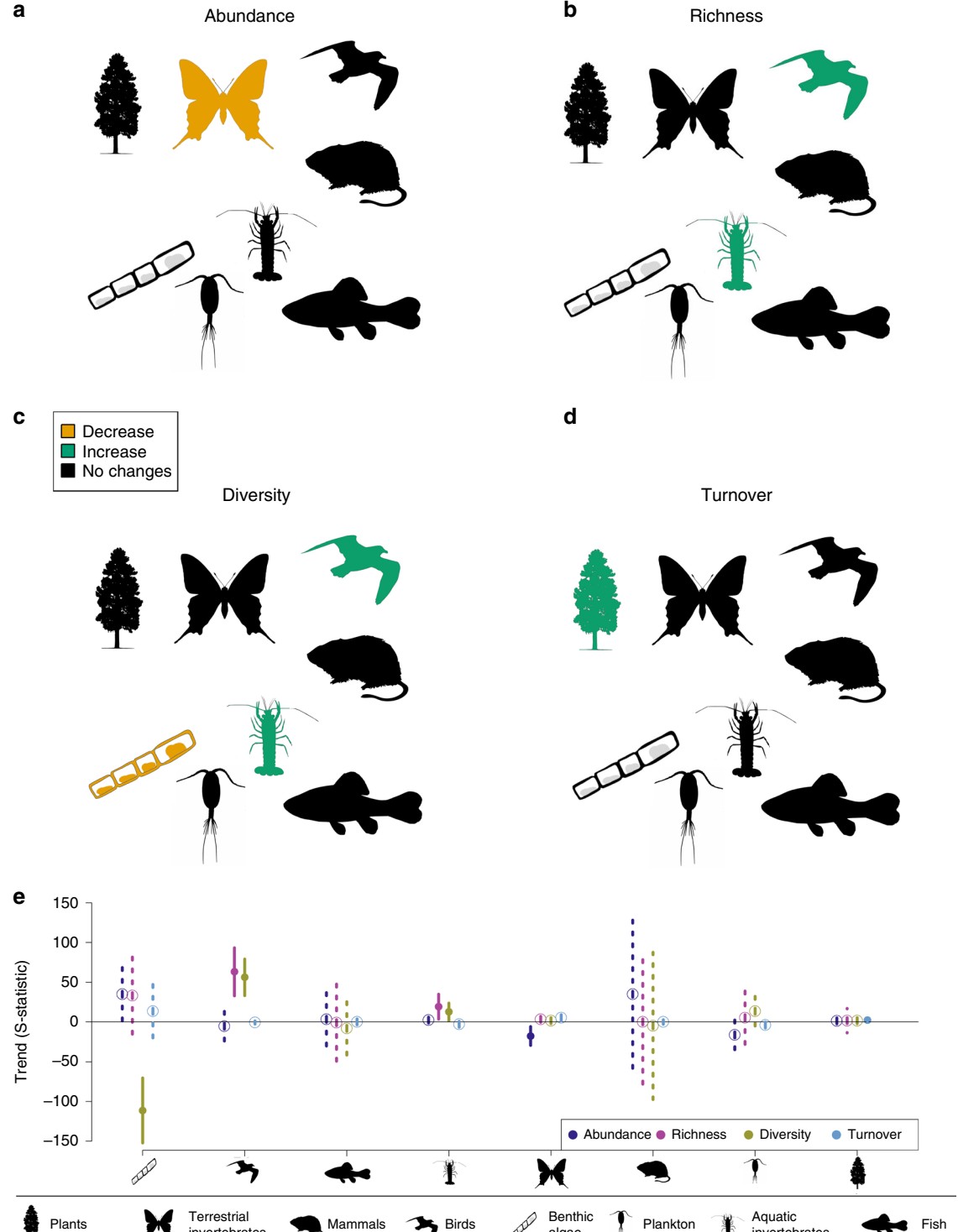

**Fig. 4 Biodiversity trends for the studied taxonomic groups.** The results of meta-analysis mixed models are shown for the four studied biodiversity metrics: abundance (**a**), richness (**b**), diversity (**c**) and turnover (**d**). Green: significant increasing trends ($p \leq 0.05$); orange: significant declining trends ($p \leq 0.05$); black: no significant trends ($p > 0.05$). Drawings from phylopic.org. **e** values of S-statistics (model estimated mean, error bar: $+/-$C.I.). Dark blue: abundance, pink: richness, yellow: diversity, light blue: turnover. Solid line and dot: $p \leq 0.05$; dashed line and open circle: $p > 0.05$. Number of time series ($n$): Plants: 34, terrestrial invertebrates: 53, mammals: 1, birds: 16, benthic algae: 7, plankton: 9, aquatic invertebrates: 38, fish: 3. Source data are provided as a Source Data file.

changes are due to direct human impact, indirect effects (see e.g., Didham et al.[47], in the context of biological invasions) or recovery processes[48].

Our model results showing positive correlations of temperature and naturalness with richness and abundance trends are consistent with other studies, e.g.[14,25]. However, we could also identify a combined effect of naturalness and temperature on abundance, richness and turnover trends. More specifically, and counterintuitive to common expectation, we found that sites considered to be in a less natural state are those experiencing the

**Table 2 Influence of climatic trends and site characteristics on biodiversity trends.**

| | Explanatory variables | z |
|---|---|---|
| Abundance | Intercept | −1.48 |
| | Temperature trend | 1.22 |
| | Precipitation trend | 0.18 |
| | Naturalness | 1.73 |
| | Latitude | −0.99 |
| | Study length | −0.30 |
| | Temperature trend: Naturalness | −2.39 |
| Richness | Intercept | 3.66 |
| | Temperature trend | 2.04 |
| | Precipitation trend | 0.38 |
| | Naturalness | 1.14 |
| | Latitude | 0.49 |
| | Longitude | 2.16 |
| | Elevation | −1.20 |
| | Study length | 0.37 |
| | Temperature trend: Naturalness | −2.02 |
| Diversity | Intercept | 1.97 |
| | Temperature trend | −0.56 |
| | Precipitation trend | −2.05 |
| | Latitude | −0.49 |
| | Longitude | 3.49 |
| | Elevation | −1.66 |
| | Study length | 1.27 |
| Turnover | Intercept | 3.35 |
| | Temperature trend | −0.60 |
| | Precipitation trend | 1.66 |
| | Naturalness | −0.52 |
| | Latitude | 1.66 |
| | Longitude | −1.21 |
| | Elevation | 4.09 |
| | Study length | 1.03 |
| | Temperature trend: Naturalness | 3.39 |

The table shows the effect sizes of the explanatory variables on the studied biodiversity metrics.

strongest changes in biodiversity metrics with increasing temperature (i.e. steeper increase in abundance and richness and steep decline in turnover). We speculate that degraded sites are more prone to invasion of generalist and invasive species, while natural sites may be more resilient[49].

A major obstacle in assessing biodiversity trends is that long-term data are unevenly distributed among taxonomic groups and are biased towards charismatic taxa, such as birds and mammals, and towards groups with long study traditions, such as vascular plants and marine fishes, while invertebrates are relatively neglected, except butterflies and bees in some countries. Our study encompasses an unusually large variety of taxonomic groups, including those largely overlooked in previous large-scale biodiversity studies[6,20]. Some of these overlooked groups that we included in our analysis, such as aquatic invertebrates, showed unexpected increases in richness and diversity, likely due to the aforementioned processes (e.g. recovery from stressors, spread of generalist or invasive taxa and taxa adapted to warmer temperatures). On the other hand, we recorded declines in species abundances for terrestrial invertebrates in the Atlantic biogeoregion, consistent with previous findings[31,32,50]. Our results therefore emphasize that patterns of change in the biodiversity of the most studied 'iconic' groups cannot be extrapolated across other taxa.

These findings reiterate the need to not only maintain but to increase the numbers of long-term monitoring schemes of local ecosystems. Such schemes can provide unique insights for ecologists and conservationists, yet they are often threatened by a lack of support as they do not fit into the temporal extent of most funding schemes. In contrast to space-for-time or aggregated snapshot (e.g. opportunistic data or atlases of species distributions) approaches, the long-term tracking of communities in specific locations minimizes the risk of biases related to shifts in sampling locations, sampling areas and protocols[51].

Our dataset is, to the best of our knowledge, the most comprehensive in terms of the spatial and temporal extent and taxonomic and realm representation of high temporal resolution long-term biodiversity monitoring records in Europe. Most sites included in this study are part of the global ILTER network and the vast majority are characterized by low anthropogenic pressure, which may have led to an underestimation of the true scale of biodiversity changes at continental scale. Perhaps most importantly, most studied sites are shielded from direct effects of changes in land-use and loss of habitat, e.g., conversion to intensive agriculture and urbanization. To overcome such sources of bias, long-term monitoring programs should include a larger representation of more intensively used (e.g. agricultural) areas and incorporate sites vulnerable to significant anthropogenic perturbations[51]. On the other hand, our approach reduces the potential risk of tracking immediate biodiversity responses to localized disturbance or successional recovery processes, which can bias the estimates of biodiversity change[52].

Although most of our time series do not predate the 1980s (only one study goes back to the 1920s), we were still able to detect evidence of substantial reorganization of communities within relatively short time frames. However, our data, as well as most of the data used in other studies[6,10,22,23], is being considered in the absence of the longer historical perspective required to capture the overall changes in biodiversity in the Anthropocene, as we lack baseline data from times when human impacts were lower (e.g. pre-industrial era). This limitation is a common issue in studies of long-term biodiversity change (but see ref. [53]) that can have a detrimental effect in restoration ecology[54,55]. The lack of baseline data could be overcome by the integration of ecological and paleobiological approaches (e.g. Battarbee et al.[56]), which could extend the temporal dimension back to, e.g., the end of the last ice age. Such approaches could have important conservation implications[57] and allow for comparisons between impacted and unimpacted sites within the same geographical area, which could reveal differences between changes driven by climate and by direct anthropogenic pressures (e.g. changes in land use or pollution)[58]. Looking into the future, there is an urgent need to harmonize biodiversity monitoring schemes[59,60] that would also allow improved up- and downscaling of trends as well as integrating cross-domain feedback loops[61].

The inherent complexity of ecological systems manifested by diverging long-term responses of local biodiversity still hamper any upscaling of these trends to a continental or even global scale. This might explain the partly contradictory results not only within but also among large-scale studies that are based on local biodiversity data[6,22]. For example, our study revealed higher taxon turnover in the terrestrial realm at European scale while, at a global scale, turnover seems higher in marine assemblages[22]. However, these studies do agree on the lack of an overall decline in species richness and on the increase in taxon turnover over time. We argue that these contradictory results can be driven by the insufficient number and quality of systemic and harmonized biodiversity monitoring activities at a local scale and by the insufficient length of the underlying time series. Regarding the latter, we should bear in mind that prior to the start of most of our biodiversity time series (mainly in the 1980s), many species had already declined in abundance or gone extinct. Moreover, the vast majority of larger scale studies describing biodiversity changes have not been able to clearly identify the environmental

drivers of these changes. This demonstrates the urgent need to complement biodiversity time series with environmental data that can be used to explain the observed patterns but are often not collected on a routine basis (with exceptions in limnology and oceanography). The complexity of biodiversity dynamics that our results and those of other studies highlight is not to be interpreted as an obstacle to the development of conservation measures. Rather we argue that a better understanding of the patterns, trends and changes in biodiversity and environmental conditions will allow conservation measures to be better tailored to specific locations and taxonomic groups in order to significantly retard or even reverse further biodiversity loss[62].

## Methods

**Data compilation**. We circulated a call for biodiversity data within the ILTER network and additional partners to fill in geographical gaps. The criteria for data selection were: (1) each time series covers at least 15 years, (2) with preferably at least ten survey events during that time, (3) sampling occurred at the same site (no space-for-time substitution), and (4) survey method, seasonal and taxonomic resolution were consistent throughout the whole study period for each time series. The final dataset included 161 time series from 89 ILTER sites and 26 additional long-term monitoring sites from other networks. The time span of the studied time series ranged between 15 and 91 years (median: 20), so that one time series started in 1921 while the others started between the 1980 and 2003 (median: 1994). The end years range between 2005 and 2018 (median: 2015; Supplementary Fig. 2).

**Biodiversity data**. Biodiversity data were expressed as abundance or biomass of surveyed taxa at each survey occasion, in some instances as percent coverage (e.g. for some time series of benthic algae and plants). Survey methods and season varied among time series, but were kept constant within each time series throughout the entire study period. In most cases, surveys were carried out once a year, but some time series had gaps or more frequent survey intervals (e.g. weekly resolution for phytoplankton, zooplankton and moths from the Finnish and Czech sites). In the latter case, we filtered the data in the time series to select only the months/seasons that were consistently surveyed throughout the whole study period, and we pooled the data (as sums or averages) within each year. Taxonomic resolution was kept constant for each time series throughout the entire study period, generally at the species or genus level, with a few exceptions (i.e. a few macroinvertebrates groups were identified at family resolution).

For each year within each time series, we computed four biodiversity metrics: total number of organisms or biomass (hereby referred to as abundance), taxonomic richness (i.e. number of taxa), Simpson´s diversity and temporal taxon turnover. The latter was computed as the proportion of taxa gained and lost between two subsequent years relative to the total number of taxa observed, using Eq. (1), as implemented in the R package codyn[63].

$$\frac{\text{Number of taxa gained} + \text{number of taxa lost}}{\text{total number of taxa}}. \qquad (1)$$

We classified the time series into nine biogeoregions, three realms and eight taxonomic groups (Fig. 1).

**Abiotic variables**. For each study site we extracted the daily mean temperature and daily total precipitation data from the gridded observational dataset for precipitation, temperature and sea level pressure in Europe (spatial resolution: 0.25 degrees[64]), and computed the mean annual temperature and total annual precipitation. For aquatic ecosystems we used in situ water temperature measured at the surface and calculated the mean temperature across the yearly monitoring periods.

We gathered information on local anthropogenic pressures in a standardized questionnaire (Supplementary Methods). The questionnaire asked the data providers to assess the impact (from 1 = no to 4 = strong impact) of a series of pressures at the site (e.g. urbanization, sources of pollution, agriculture, etc.; for a full list see Supplementary Methods) and indicate whether impacts were constant or changed throughout the study period. Data providers were also asked to estimate the overall environmental quality of the site (i.e. naturalness; from 1 = low to 5 = high) and to state whether it was constant or changed throughout the study period. We used this information to define the quality-class of the sites, based on the overall assessment. The majority of sites (n = 67) scored 5 (i.e. high naturalness), 42 sites scored 4, 37 sites scored 3 and 15 sites scored 2.

**Data analysis**. We used a two-step procedure that allowed us to combine very heterogeneous original datasets (see paragraph Biodiversity data). First, we analyzed each time series separately to quantify time-series-specific biodiversity trends. Second, we used the effect sizes of the individual time-series-specific biodiversity trends to synthesize the overall trends and identify common patterns and drivers. The results of this second (meta-)analysis are reported in the paper.

For the first step of the analysis, we used the Mann–Kendall trend test to identify monotonic trends in each biodiversity and climate time series over the study period[65,66]. We detected serially correlated time series using auto- and cross-covariance and correlation functions[67], and we applied the modified Mann–Kendall with Hamed and Rao[68] variance correction approach. We used S-statistic and its variance as effect size of the trend[65] for the next step of the analysis. A similar meta-analytical approach has already been applied in a previous study on ecological time series[69].

At 23 study sites, multiple time series were available for a given taxonomic group, e.g., when surveys were conducted at multiple transects or plots or with multiple traps. To avoid pseudoreplication, we combined those time series using meta-analysis mixed models (using the R package metafor[70]) and extracted the cumulative effect sizes and their variances prior to the second step of the analysis. The total of 161 time series reported above refers to the aggregated final set of time series.

The second step of the analysis aimed at synthesizing the trends across the different time series. For that, we fitted meta-analysis mixed models to account for random effects. To compute the overall biodiversity trends and to explore how the trends varied among biogeoregions, realms and taxonomic groups, we included biogeoregion (nine levels), realm (three levels: freshwater, marine and transitional zone, and terrestrial) and taxonomic group (eight levels) as explanatory variables in the models with no intercept. We did so separately because biogeoregion, realm and taxonomic group were not independent (see Supplementary Table 1). The results of these models hence show whether or not the overall S-statistics for individual trends in the groups differ significantly from zero.

Additional meta-analysis mixed models were used to test the influence of selected abiotic variables describing site characteristics and climate on the biodiversity trends. We included the following variables: latitude, longitude, elevation, site naturalness, S-statistics[65] of temperature and precipitation trends, and the length of each time series. These explanatory variables showed little collinearity ($|r| < 0.6$), and were thus all retained as potential predictors. Similar to Everaert et al.[71], we applied an information-theoretic approach to model selection and multi-model inference[72], to determine the relative importance of those explanatory variables on the trends in biodiversity metrics. For this we created a candidate set of models with all possible linear combinations of explanatory variables and we extracted the corrected Akaike's information criterion (AICc) of each model using the R package glmulti[73]. We retained only plausible models with δAICc ≤ 2 of the best model (i.e. the one(s) with the lowest AICc[72]) and computed the relative importance of each predictor variable as the sum of the Akaike's weights of all the selected models in which that variable was included. We computed the model-averaged coefficients (±95% C.I.) for each predictor variable in each selected model, weighted by the Akaike's weights (Supplementary Table 2). To evaluate the effect of the interaction among climatic and local stressors, we added to the selected models the interactions between site naturalness and temperature trends, and the interaction between site naturalness and precipitation trends. We then compared the resulting models (without interaction, with single interaction, with both interactions) and chose the one with the lowest AIC value. We have implemented this procedure because, to the best of our knowledge, it is not possible to include selected interactions into the information-theoretic approach to model selection and including all interactions would have resulted in an overly complex model.

To account for biases in biogeoregions, realms and taxonomic groups, we also performed sensitivity analyses (Supplementary notes and Supplementary Table 3), which confirmed that the results were robust.

**Reporting summary**. Further information on research design is available in the Nature Research Reporting Summary linked to this article.

## Data availability

All datasets analyzed during the current study have been deposited in public repositories, at the links reported in Supplementary Data. Please note that some of the datasets will be publicly accessible after a period of embargo, from 1st January 2021. The source data underlying all figures are provided as a Source Data file. Source data are provided with this paper.

## Code availability

The R code used for all analyses is available at the GitHub repository: https://github.com/Biodiversity-trends-in-Europe-ILTER/R-code. Source data are provided with this paper.

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

## Acknowledgements

We are grateful to the ILTER network and the eLTER PLUS project (Grand Agreement No. 871128) for financial support. We acknowledge the E-OBS dataset from the EU-FP6 project ENSEMBLES (http://ensembles-eu.metoffice.com) and the data providers in the ECA&D project (http://www.ecad.eu). The evaluation of forest plant diversity was based on data collected by partners of the official UNECE ICP Forests Network (http://icp-forests.net/contributors); part of the data were co-financed by the European Commission, project LIFE 07 ENV/D/000218 "Further Development and Implementation of an EU-level Forest monitoring Systeme (FutMon)". Data on wintering water birds in Bulgaria were provided by the national Executive Environment Agency with the Ministry of Environment and Waters. Data from the Finnish moth monitoring scheme were supported by the Finnish Ministry of the Environment. Data from the Swedish ICP Integrated Monitoring sites were financed by the Swedish Environmental Protection Agency. Data collection at Esthwaite Water and a subset of UK ECN sites was supported by Natural Environment Research Council award number NE/R016429/1 as part of the UK-SCaPE programme delivering National Capability. Sponsorship of other UK ECN sites contributing data was provided by Agri-Food and Biosciences Institute, Biotechnology and Biological Sciences Research Council, Department of Environment Food and Rural Affairs, Natural Resources Wales, Defense Science Technology Laboratory, Environment Agency, Forestry Commission, Forest Research, the James Hutton Institute (The Rural & Environment Science & Analytical Services Division of the Scottish Government), Natural England, Rothamsted Research, Scottish Government, Scottish Natural Heritage and the Welsh Government. Data from the Mondego estuary (Portugal) were supported by the Centre for Functional Ecology Strategic Project (UID/BIA/04004/2019) within the PT2020 Partnership Agreement and COMPETE 2020, and by FEDER through the project ReNATURE (Centro 2020, Centro-01-765-0145-FEDER-000007). We would like to thank Limburgse Koepel voor Natuurstudie (LiKoNa) for the data related to the National Park Hoge Kempen (BE). We would like to acknowledge the support for the long-term monitoring program MONEOS in the Scheldt estuary (BE) by 'De Vlaamse Waterweg' and 'Maritieme Toegang' (Flemish government). We are grateful to the board of the National Park "De Hoge Veluwe" for the permission to conduct our research on their property. We thank Ian J. Winfield and Terje Bongard for contributing data for the sites: Bassenthwaite Lake, Derwent Water (UK) and Atna River (Norway, freshwater invertebrate time series). Open access funding provided by Umeå University.

## Author contributions

F.P. and P.H. conceived the study. F.P. analyzed the data, with inputs by I.K. F.P. wrote the manuscript, with major contributions by P.H. and I.K. R. Adrian, R. Alber, A.A., C. A., J.B., L.B., D.B., N.B., S.B., D.S.B., V.B., E.C., R.C., P.C., B.J.E., G.E., V.E., H.F., R.G.G., D.G.G., U.G., J.M.G., L. Hadar, L. Halada, M.H., H.H., K.L.H., B.J., T.C.J., H.K., I.K.S., I. K., R.L., F.M., H.M., J.M., S.M., D.M., B.P.N., D. Oro, D. Ozoliņš, B.M.P., D.P., M.P., M. Â.P., B.P., T.P., J.P., S.M.S., M.S., S.C.S., A.S., K.S., G.S., R.S., J.A.S., S.S., L.S., A.T., G.V. H., G.V.R., M.E.V., S.V. and P.H. provided data and contributed to the different versions of the manuscript.

## Competing interests

The authors declare no competing interests.

## Additional information

Francesca Pilotto [1,2 ✉], Ingolf Kühn [3,4,5], Rita Adrian[6], Renate Alber[7], Audrey Alignier[8,9], Christopher Andrews [10], Jaana Bäck[11], Luc Barbaro [12], Deborah Beaumont[13], Natalie Beenaerts [14], Sue Benham [15], David S. Boukal [16,17], Vincent Bretagnolle[18,19], Elisa Camatti [20], Roberto Canullo [21], Patricia G. Cardoso[22], Bruno J. Ens [23], Gert Everaert [24], Vesela Evtimova [25], Heidrun Feuchtmayr [26], Ricardo García-González[27], Daniel Gómez García[27], Ulf Grandin [28], Jerzy M. Gutowski [29], Liat Hadar[30], Lubos Halada [31], Melinda Halassy [32], Herman Hummel [33], Kaisa-Leena Huttunen[34,35], Bogdan Jaroszewicz [36], Thomas C. Jensen [37], Henrik Kalivoda[38], Inger Kappel Schmidt [39], Ingrid Kröncke[40], Reima Leinonen[41], Filipe Martinho [42], Henning Meesenburg [43], Julia Meyer[40], Stefano Minerbi[44], Don Monteith [26], Boris P. Nikolov [25], Daniel Oro[45,46], Dāvis Ozoliņš[47],

Bachisio M. Padedda [48], Denise Pallett[49], Marco Pansera[20], Miguel Ângelo Pardal[42], Bruno Petriccione [50], Tanja Pipan [51], Juha Pöyry [52], Stefanie M. Schäfer[49], Marcus Schaub [53], Susanne C. Schneider[54], Agnija Skuja[47], Karline Soetaert[33], Gunta Spriņģe[47], Radoslav Stanchev[25], Jenni A. Stockan[55], Stefan Stoll[56,57], Lisa Sundqvist[58], Anne Thimonier [53], Gert Van Hoey [59], Gunther Van Ryckegem [60], Marcel E. Visser [61], Samuel Vorhauser[7] & Peter Haase [1,57✉]

[1]Senckenberg Research Institute and Natural History Museum Frankfurt, Gelnhausen, Germany. [2]Environmental Archaeology Lab, Department of Historical, Philosophical and Religious Studies, Umeå University, Umeå, Sweden. [3]Department of Community Ecology, Helmholtz Centre for Environmental Research - UFZ, Halle, Germany. [4]Martin Luther University Halle-Wittenberg, Geobotany and Botanical Garden, Halle, Germany. [5]German Centre for Integrative Biodiversity Research (iDiv) Halle - Jena - Leipzig, Leipzig, Germany. [6]Department of Ecosystem Research, Leibniz Institute of Freshwater Ecology and Inland Fisheries & Department of Biology, Chemistry and Pharmacy, Freie Universität Berlin, Berlin, Germany. [7]Biological Laboratory, Agency for Environment and Climate Protection, Bolzano, Italy. [8]UMR 0980 BAGAP, INRAE – Institut Agro – ESA, Rennes, France. [9]LTSER Zone Atelier Armorique, 35042 Rennes, France. [10]UK Centre for Ecology & Hydrology, Bush Estate, Penicuik, Midlothian, UK. [11]Institute for Atmospheric and Earth system Research, Department of Forest Sciences, University of Helsinki, Helsinki, Finland. [12]Dynafor, INRAE, University of Toulouse, France & CESCO, Muséum National d'Histoire Naturelle, Sorbonne-Univ, Paris, France & LTSER Zone Atelier Pyrénées Garonne, Auzeville-Tolosane, France. [13]Rothamsted Research, North Wyke, Okehampton, Devon, UK. [14]Centre for Environmental Sciences, Hasselt University, Hasselt, Belgium. [15]Forest Research, Farnham, UK. [16]University of South Bohemia, Faculty of Science, Department of Ecosystem Biology & Soil and Water Research Infrastructure, Ceske Budejovice, Czech Republic. [17]Czech Academy of Sciences, Biology Centre, Institute of Entomology, Ceske Budejovice, Czech Republic. [18]CEBC, UMR7372, CNRS & La Rochelle University, 79360 Villiers en bois, France. [19]LTSER Zone Atelier Plaine & Val de Sèvre, 79360 Beauvoir sur Niort, France. [20]Institute of Marine Sciences, National Research Council, Venice, Italy. [21]School of Biosciences and Veterinary Medicine, unit Plant Diversity and Ecosystems Management, University of Camerino, Camerino, Italy. [22]CIIMAR, Interdisciplinary Centre of Marine and Environmental Research of the University of Porto, Porto, Portugal. [23]Sovon Dutch Centre for Field Ornithology, Nijmegen, The Netherlands. [24]Flanders Marine Institute, Ostend, Belgium. [25]Institute of Biodiversity and Ecosystem Research, Bulgarian Academy of Sciences, Sofia, Bulgaria. [26]UK Centre for Ecology & Hydrology, Lancaster Environment Centre, Lancaster, UK. [27]Instituto Pirenaico de Ecología (CSIC), Jaca, Spain. [28]Department of Aquatic Sciences and Assessment, Swedish University of Agricultural Sciences, Uppsala, Sweden. [29]Department of Natural Forests, Forest Research Institute, Białowieża, Poland. [30]Ramat Hanadiv, Zikhron Ya'akov, Israel. [31]Institute of Landscape Ecology SAS, Branch Nitra, Slovakia. [32]MTA Centre for Ecological Research, Institute of Ecology and Botany, Vácrátót, Hungary. [33]Royal Netherlands Institute for Sea Research, and Utrecht University, Yerseke, The Netherlands. [34]Department of Ecology and Genetics, University of Oulu, Oulu, Finland. [35]Oulanka Research Station, University of Oulu Infrastructure Platform, Kuusamo, Finland. [36]Białowieża Geobotanical Station, Faculty of Biology, University of Warsaw, Białowieża, Poland. [37]Norwegian Institute for Nature Research, NINA-Oslo, Norway. [38]Institute of Landscape Ecology SAS, Bratislava, Slovakia. [39]Geosciences and Natural Resource Management, University of Copenhagen, Copenhagen, Denmark. [40]Senckenberg am Meer, Marine Research Department, Wilhelmshaven, Germany. [41]Kainuu Centre for Economic Development, Transport and the Environment, Kajaani, Finland. [42]Centre For Functional Ecology (CFE), Department of Life Sciences, University of Coimbra, Coimbra, Portugal. [43]Northwest German Forest Research Institute, Göttingen, Germany. [44]Forest Services, Autonomous Province of Bolzano - South Tyrol, Bolzano, Italy. [45]CEAB (CSIC), 17300 Blanes, Spain. [46]IMEDEA (CSIC-UIB), 07190 Esporles, Spain. [47]Institute of Biology, University of Latvia, Salaspils, Latvia. [48]Dipartimento di Architettura, Design e Urbanistica, Università degli Studi di Sassari, Sassari, Italy. [49]UK Centre for Ecology & Hydrology, Wallingford, UK. [50]Carabinieri, Biodiversity and Park Protection Department, Castel di Sangro Biodiversity Unit, L'Aquila, Italy. [51]ZRC SAZU Karst Research Institute, Ljubljana & UNESCO Chair on Karst Education University of Nova Gorica, Vipava, Slovenia. [52]Finnish Environment Institute (SYKE), Biodiversity Centre, Helsinki, Finland. [53]Swiss Federal Institute for Forest Snow and Landscape Research WSL, Birmensdorf, Switzerland. [54]Norwegian Institute for Water Research, Oslo, Norway. [55]Ecological Sciences, James Hutton Institute, Craigiebuckler, Aberdeen, UK. [56]University of Applied Sciences Trier, Environmental Campus Birkenfeld, Birkenfeld, Germany. [57]University of Duisburg-Essen, Essen, Germany. [58]Swedish Meteorological and Hydrological Institute, Gothenburg, Sweden. [59]Flanders Research Institute for Agriculture, Fishery and Food, Oostende, Belgium. [60]Research Institute for Nature and Forest, Brussels, Belgium. [61]Department of Animal Ecology, Netherlands Institute of Ecology (NIOO-KNAW), Wageningen, The Netherlands. ✉email: francesca.pilotto@umu.se; peter.haase@senckenberg.de

