## [Peer Review File · Nature Communications]

Reviewers' Comments:

Reviewer #1:

Remarks to the Author:

The study "Changes in long-term biodiversity trends in Europe" meta-analyzes time-series data on different taxa from different environments across Europe. It represents a very timely and relevant contribution to the ongoing debate about biodiversity loss across scales. Particularly relevant is the question regarding the generality of trends. It also highlights the outstanding importance of long-term monitoring as done by the LTER network. I think the study is highly suitable for Nature Communications.

Still, I have some comments list according to their appearance in the text. Most important are two issues: 1) the issue of land use, whether different land-use types can be compared and if not because only few were included, the need to point out this restriction early on. 2) Some discussion should be added regarding conservation to avoid that the lack of generality is used as excuse for doing nothing. For more details, see below:

What I miss throughout the Abstract and Introduction (e.g. L79, L95, L133) is that different drivers of declines may interact (e.g. land use, climate change, nitrogen deposition,...) and that these drivers may differ geographically leading to different trends in different regions or in different taxa;

L132: community stability and turnover should be defined briefly when mentioned first

L219: site naturalness should be defined briefly when mentioned first

L285: I am not very familiar with aquatic systems, but hasn't water quality increased over the past decades in many areas, e.g. Central Europe? If so, the positive trends for some freshwater groups may not be so surprising

L302: It should be made clear early in the paper which land-use types are covered. I suggest an additional analysis comparing trends between coarse terrestrial land-use categories (e.g. forest, grassland, arable fields, ...). This links also to my comment above about the importance of different drivers, particularly land use, which is an important part of the ongoing discussion (see e.g. Bowler et al 2019 Conservation Biology, Seibold et al 2019 Nature)

L322: please rephrase and specify since climate change is also anthropogenic

L340: What I miss in the conclusion is some kind of cautionary note regarding the relevance towards measures to halt biodiversity losses. The complexity of overall trends and differences between regions and taxa are a very important finding, but could be easily abused to say that scientists do not agree whether there are declines or not and thus, there is no need to take costly actions. In my opinion, there is enough evidence that e.g. terrestrial invertebrates have declines which is also confirmed in by this study at least for some regions.

L381: I am a bit surprised to find only pressures such as recreation or urbanization mentioned here, since the former is, to my knowledge, hardly considered a major factor and the later seems to be rather irrelevant for the datasets used here (L302). Please provide the whole list of pressure and clarify the reasoning. Repeating my comment above, I think the issue of land-use types should be addressed more clearly.

L385: The scoring of naturalness should be specified. Please provide also some examples, within and between land-use types.

FigS1: please point out in the main text or methods that only one study goes back to 1920 and all

other started around 1980

Reviewer #2:

Remarks to the Author:

Please find the attached pdf file.

Reviewer #3:

Remarks to the Author:

This paper reports temporal biodiversity trends in 161 time series for various habitats, regions, and taxa in Europe, most of a duration of 20 years or so. The goal is to help advance our understanding of local biodiversity trends and their causes, which have been the focus of considerable recent debate. More data is always welcome on a topic of general interest, so this study is certainly a worthwhile contribution. That said, the nature of the data and conclusions in this manuscript do not add up to what I would consider a major advance (I elaborate on this point below).

(1) The overarching conclusions and data limitations are basically the same as in previous studies, the latter of which used more data from a broader range of systems. In terms of data, it is true that some other individual studies either had less data or a more restrictive taxonomic focus, but taken together they cover a broader range of situations and tell essentially the same story: diversity (usually estimated as richness) shows little overall trend over time or a weak positive trend, on average, while there is considerable species turnover, and some particular circumstances (e.g., a particular taxon in a particular habitat) that deviate from the overall trends. The recent Blowes et al. paper in *Science* covers many taxa and regions across the globe with the same conclusion. European data sets (the focus of the present study) are particularly well represented in previous analyses. None of this means that the study under review is not a meaningful contribution, but these considerations might be relevant to the editor in deciding whether *Nature Communications* is the right venue.

(2) In terms of data limitations, the data here are of essentially the same nature as in *BioTime* (Blowes et al. and Dornelas et al.), and so they do not overcome previously noted limitations (doing so would more clearly represent an advance). While there are many regions, habitats, and taxa represented, the representation of combinations is actually fairly limited. Table S2 shows that roughly 17% of bioregion-realm-taxon combinations are represented, and less than 10% are represented by more than 1 study. This manifests clearly in the Discussion, where differences among regions or taxa are often interpreted with reference to a different variable (e.g., a pattern in a particular region is likely because that region only has data sets for particular taxa). Perhaps the most glaring example (line 209) is algae being represented by one river catchment, which probably means there should be no category "algae" in this analysis. So, while we can see that there is heterogeneity among time series, there is limited scope for drawing strong conclusions about trends for particular taxa or regions (i.e., what factors underlie the heterogeneity). As such, when the authors say (line 123) that "a comprehensive overview directly comparing trends among complementary datasets has been lacking so far", it's not so clear that this "lack" has been filled by the current study.

(3) In revising the manuscript, I think clarity could be improved with greater precision of several statements or arguments. For example:

(a) Community stability. This concept is emphasized in the abstract, but in the Methods it is revealed that this just means species turnover (specifically the inverse). So, a community in which composition is changing is considered "unstable". Furthermore, species turnover is interpreted as "an early warning of more severe biodiversity changes to come". I can't think of any basis in the

data for this. When environmental conditions change, fitness declines for some species and increases for others, such that turnover may well be a sign of adaptation (in the broad sense). I think a less value-laden presentation would simply call it turnover. Presenting possible interpretations in the Discussion is reasonable, but I see no basis for assuming there is any indication of "instability" or a "warning" of bad things to come.

(b) Line 259: This line is hard to follow. "extended on a larger geographic area". How so?

(c) Line 263: "turnover driven changes in community stability". This doesn't really make sense, since stability is defined as low turnover. One therefore can't "drive" the other (one is the other). The subsequent list of explanations is also difficult to follow: e.g., "stochastic responses to climate change" (stochastic in what sense?) On line 266, it is not clear how compensatory dynamics is a different explanation from the others (one could be embedded in the other).

(d) Line 318: "a detrimental effect in restoring actual trends in biodiversity". Not clear what is meant by this.

(e) Line 253: What is meant by the word "masked"? One set of species declining and another increasing are two different things. If one looks specifically for either, they can find it. If they look at something else (e.g., net change in the number of species), they will not. But nothing is being "masked".

(f) There is ambiguity with respect to whether temperature is treated as an average or a trend (it says different things in different places).

Responses to Reviewer Comments

Responses to reviewer comments are written in italicized blue text under each comment.

Reviewer #1 (Remarks to the Author):

The study “Changes in long-term biodiversity trends in Europe” meta-analyzes time-series data on different taxa from different environments across Europe. It represents a very timely and relevant contribution to the ongoing debate about biodiversity loss across scales. Particularly relevant is the question regarding the generality of trends. It also highlights the outstanding importance of long-term monitoring as done by the LTER network. I think the study is highly suitable for Nature Communications.

Response #1: We thank the reviewer for the positive comments regarding the manuscript.

Still, I have some comments list according to their appearance in the text. Most important are two issues: 1) the issue of land use, whether different land-use types can be compared and if not because only few were included, the need to point out this restriction early on.

Response #2: No specific land-use types were included in the analyses. Each site was assessed by the data providers based on its naturalness and potential anthropogenic impacts (e.g. urbanization, agriculture or pasture, hunting) during the period the relevant time series was studied. Appendix B reports the information gathered by the data providers.

2) Some discussion should be added regarding conservation to avoid that the lack of generality is used as excuse for doing nothing.

Response #3: We agree with the reviewer that the “lack of generality” can pose a challenge for nature conservation. This issue has already been included in the Introduction (lines 112- 114): “Therefore, it is crucial to understand how often and how strongly local biodiversity trends contrast and why they vary across biota, regions and local conditions²⁰.” We further discuss this issue in the Discussion, at lines 336-337 “The inherent complexity of ecological systems manifested by diverging long-term responses of local biodiversity still hamper any upscaling of these trends to a continental or even global scale.”; and at lines 351-356: “The complexity of biodiversity dynamics that our results and those of other studies highlight is not to be interpreted as an obstacle to the development of conservation measures. Rather we argue that a better understanding of the patterns, trends and changes in both biodiversity and environmental conditions, will allow developing conservation measures that are tailored to the specific local context of the target sites and taxonomic groups and that will eventually bring the biodiversity loss to a halt⁶²”.

For more details, see below:

What I miss throughout the Abstract and Introduction (e.g. L79, L95, L133) is that different drivers of declines may interact (e.g. land use, climate change, nitrogen deposition,...) and that these drivers may differ geographically leading to different trends in different regions or in different taxa;

Response #4: We agree with the reviewer that different drivers likely interact in their effects on biodiversity. We have now included the interaction between site naturalness (a composite measure of local anthropogenic pressure) and temperature and precipitation trends in our analysis. Accordingly, we updated the section of the Methods (lines 448-455), Results (lines 224-236 and Table 2) and Discussion (lines 280-286), and we added Figure S1 (Supplementary material) to aid the interpretation of those interactions.

L132: community stability and turnover should be defined briefly when mentioned first

Response #5: We specify that we refer to community TEMPORAL stability at line 97 (Abstract), at line 124, 146 and 154 (Introduction), and at line 383 (Methods). We added the definition of taxon turnover at line 125: "..., i.e. a measure of temporal community variability" and we added the equation that we used to compute it (Method section, line 383-397).

L219: site naturalness should be defined briefly when mentioned first

Response #6: We added the definition of site naturalness at line 225 : "(i.e. a measure of local anthropogenic pressure)"

L285: I am not very familiar with aquatic systems, but hasn't water quality increased over the past decades in many areas, e.g. Central Europe? If so, the positive trends for some freshwater groups may not be so surprising

Response #7: Aquatic systems in Central Europe have improved since the '60s - '70s , mainly due to waste water treatment, limitation to the use of DDT and decreases in acid rain (see Van Looy et al. 2016, Vaughan & Ormerod 2012). Over the last decade, however, their quality has, overall, not changed much, and most of them remain under significant pressure from diffuse (e.g. agriculture) and point-source (e.g. industry) pollution, over-abstraction and hydromorphological changes stemming from a range of human activities (see Report from the commission to the European Parliament and the Council on the implementation of the water framework Directive and the Floods Directive, as well as The European Environment Agency report European waters – Assessment of status and pressures 2018).

However, most aquatic sites within the manuscript originate from boreal biogeoregions where freshwater systems seem to recover from acidification and hydromorphological stress. We discuss this point at lines 269-272: "Over the past 30-40 years (i.e. the period covered by most time series in our dataset) levels of air pollution, especially acid rain and nitrogen emissions, declined substantially, which could be an important driver of biotic change (see e.g. Monteith et al. ⁴⁵).". We have added this sentence to line 295-296 to make it clearer: "likely due to an aforementioned recovery process in response to decreased stressors."

Cited literature:

Van Looy et al. 2016. Long-term changes in temperate stream invertebrate communities reveal a synchronous trophic amplification at the turn of the millennium. Sci. Total Environ. 565, 481–488.

Vaughan & Ormerod 2012. Large-scale, long-term trends in British river macroinvertebrates. Glob. Chang. Biol. 18, 2184–2194. <https://doi.org/10.1111/j.1365-2486.2012.02662.x>.

L302: It should be made clear early in the paper which land-use types are covered. I suggest an additional analysis comparing trends between coarse terrestrial land-use categories (e.g. forest, grassland, arable fields, ...). This links also to my comment above about the importance of different drivers, particularly land use, which is an important part of the ongoing discussion (see e.g. Bowler et al 2019 Conservation Biology, Seibold et al 2019 Nature)

Response #8: We fully agree that land use is an important factor affecting local biodiversity trends. However, it was impossible to include it as a driver per se, as the time series we have studied span 15-92 years and no such data (i.e., more or less continuous time series) were available for land-use types. We tried to compensate for this by using “site naturalness”, as a synthetic measure of anthropogenic impacts at the sites,

L322: please rephrase and specify since climate change is also anthropogenic

Response #10: Rephrased as “changes driven by climate and by direct anthropogenic pressures (e.g. changes in land use or pollution)”, at line 332.

L340: What I miss in the conclusion is some kind of cautionary note regarding the relevance towards measures to halt biodiversity losses. The complexity of overall trends and differences between regions and taxa are a very important finding, but could be easily abused to say that scientists do not agree whether there are declines or not and thus, there is no need to take costly actions. In my opinion, there is enough evidence that e.g. terrestrial invertebrates have declines which is also confirmed in by this study at least for some regions.

Response #11: We changed the final sentence of the Discussion to put more emphasis on the need for further conservation measures (lines 351-356): “The complexity of biodiversity dynamics that our results and those of other studies highlight is not to be interpreted as an obstacle to the development of conservation measures. Rather we argue that a better understanding of the patterns, trends and changes in both biodiversity and environmental conditions, will allow developing conservation measures that are tailored to the specific local context of the target sites and taxonomic groups and that will eventually bring the biodiversity loss to a halt⁶².”

L381: I am a bit surprised to find only pressures such as recreation or urbanization mentioned here, since the former is, to my knowledge, hardly considered a major factor and the later seems to be rather irrelevant for the datasets used here (L302). Please provide the whole list of pressure and clarify the reasoning. Repeating my comment above, I think the issue of land-use types should be addressed more clearly.

Response #12: Those reported in the text were only examples of the pressures that were taken into account. We have now changed it to a more representative list: “ e.g. urbanization, sources of pollution, agriculture, etc.; for a full list see Appendix B” (lines 399-400). The whole list of pressures is available in the survey used to collect information on individual sites (Appendix B).

L385: The scoring of naturalness should be specified. Please provide also some examples, within and between land-use types.

Response #13: The scoring system is specified as ranging from 1=low to 5=high naturalness (lines 402). Land-use types have not been directly used to estimate naturalness (as explained above in Responses #2, 4 and 8).

FigS1: please point out in the main text or methods that only one study goes back to 1920 and all other started around 1980

Response #14: We specify this at lines 320-321 (Discussion): “Although most of our time series do not predate the early 1990s (only one study goes back to the 1920s),...”; and at line 365-369 (Methods): “The time span of the studied time series ranged between 15 and 92 years (median: 21), so that one time series started in 1921 while the others started between the 1980s and 2003 (median: 1993). The end years range between 2005 and 2018 (median: 2015; Supplementary Fig. S2).”.

Reviewer #2 (Remarks to the Author):

Review report on "Changes in long-term biodiversity trends in Europe" by Francesca Pilotto et al.

This manuscript analysed 161 abundance/biomass time series of 15–92 year time span across Europe, asking three key questions:

1. whether a long-term trend in biodiversity is detectable at each location;
2. how the trends attribute to climate changes and/or changes in local conditions;
3. whether the observed trends vary amongst bioregions, realms and taxonomic groups.

The authors have found and reported that the direction of long-term trends varies depending upon bioregions, realms and taxonomic groups. The changing patterns in biodiversity are complex.

I appreciate the authors' hard work in compiling the data, forming the research network from different regions and expertise. However, it would be more beneficial for the reader if the authors could convey a more transparent and informative message, highlighting the novelty of the work. At the moment, I feel that some parts of ambiguity conceal the novelty of the paper, which is a shame, understanding the authors' great efforts. I am going to provide my general

comments below, focusing on the statistical logic used, with the hope that the authors find them useful for further consideration.

Response #15: We thank the reviewer very much for his/her comments. We believe that there is a difference between ambiguity of results (or, as in our case, heterogeneity rather than ambiguity) and novelty. We also do not believe that only 'crystal clear' (i.e. simple) results can provide novelty, especially in studies covering multiple taxa and ecosystems over large spatial scales.

To the best of our knowledge, the only analysis coming up with a similar story is that of Blowes et al. (2019) Science, 366, 339-345, which just came out after we submitted our manuscript. In contrast to Blowes et al., our study has a much higher temporal resolution and is indeed based on site-specific trends and not on biodiversity studies repeated in a similar region (i.e. 96-km² hexagonal cells). Additionally, and in contrast to Blowes et al., we included some independent variables (site naturalness, temperature, precipitation...) in our analyses to better understand the observed trends in biodiversity metrics. We now highlight the novelty better in the Introduction (Lines 118-137). Please also see response #21 below.

General comments

I feel that overall the logic of the paper was heavily dominated by the significance of statistical hypothesis tests, whether the long-term trend was significant or not, and the logic conceals the novelty of the paper.

Response #16: We have removed the results of statistical hypothesis testing from the main text into a table. We have also streamlined the results section accordingly to tell the story in a more concise way. We believe that the revised version better explains the main patterns and highlights the novelty of our study without being distracted by the statistical details in the text.

First, the authors claim that there were no significant trends (no changes) found in a large majority of bioregions, realms and taxonomic groups. However, such a statement sounds a little bit strong. Since the authors failed to reject the null hypothesis that is no monotonic trend in a time series, its interpretation ought to be no conclusion, where there exists a trend or not, for those regions, realms and groups, rather than "no change was found". This forces the authors in a passive mode, rather than actively stating the results, and induces ambiguity, whether the amplitude of the long-term trends is overall small or fluctuates largely in those regions, realms and groups.

Response #17: We are afraid that there was a misunderstanding here. The results presented are the ones from the meta-analysis mixed-effect models, not those of the non-parametric trend analyses. We only used S statistics and variances for the meta-analytical approach to synthesize the heterogeneous datasets into a common statistical framework. Therefore, the results show whether in the respective groups the S statistics of each individual time series differ significantly from zero. This indeed means that an individual time series might have a positive, and another in the same group a negative trend. Our aim is not to show the individual

trends, but the overall trend in a group. Our aim hence is to show the overall results as presented in Figures 2-4, rather than the underlying individual results as in the respective appendices.

Second, I wonder what the statistical power, the probability of failing to reject the null hypothesis when it is actually false, for the present test would be. From Fig S1, I suppose that the majority of time series possess the time length about 20–30 years. Reflecting the fact that the test used is non parametric one, I imagine that the power would not be that high. There is, of course, no doubt that 20–30 year time series records are precious ones although, whether it is long enough to provide a enough statistical power is a different question.

Response #18: As stated above, the shown results are not those of the non-parametric Mann-Kendall trend test, and the decreased power of the non-parametric test does not influence the error probability which was computed comparing these coefficients in the meta-analysis mixed-effect models. We believe that we explain this approach much better in the revised version. Applying the additional sensitivity tests makes our results generalizable and robust. Our analysis hence does not suffer from the problem the reviewer discussed, based on the misunderstanding indicated above.

Combining, I wonder if it is wise to push the statistical hypothesis testing results as in the current stand, to convey the key message. I am quite sure that the authors could provide more informative results and figures, rather than the current figures provided in the main part of the manuscript. For example, showing the distribution of slope coefficients or equivalent parameters that summarise monotonic trends instead could be more informative, and I believe that the reader would more appreciate it. In this regard, I have found figures S2–4 more informative. I would like to encourage the authors for further consideration.

Response #19: As mentioned above (our response #17), our aim is to show the overall trend in a group as presented in Figures 2-4. However, we see the point raised by the reviewer, so we added a panel to Figures 2-4 ,providing additional quantitative information (in line with figures S2–4 of the previous submission).

Reviewer #3 (Remarks to the Author):

This paper reports temporal biodiversity trends in 161 time series for various habitats, regions, and taxa in Europe, most of a duration of 20 years or so. The goal is to help advance our understanding of local biodiversity trends and their causes, which have been the focus of considerable recent debate. More data is always welcome on a topic of general interest, so this study is certainly a worthwhile contribution. That said, the nature of the data and conclusions in this manuscript do not add up to what I would consider a major advance (I elaborate on this point below).

Response #20: Regarding “the nature of the data and conclusions” we believe that there are considerable advances on which we will expand in more detail further down. Nevertheless, to

the best of our current knowledge, only two studies have been published that analyzed a larger number of biodiversity time series covering different taxonomic groups at larger spatial scales (continental or global) which are Dornelas et al. 2014 and Blowes et al. 2019 (both in Science).

As Maria Dornelas is either first or senior author in the two aforementioned studies, we contacted her with regard to the first two issues raised by the reviewer (see below). In particular, she confirmed the significant differences in the data used in our study to that of the two other studies.

(1) The overarching conclusions and data limitations are basically the same as in previous studies, the latter of which used more data from a broader range of systems. In terms of data, it is true that some other individual studies either had less data or a more restrictive taxonomic focus, but taken together they cover a broader range of situations and tell essentially the same story: diversity (usually estimated as richness) shows little overall trend over time or a weak positive trend, on average, while there is considerable species turnover, and some particular circumstances (e.g., a particular taxon in a particular habitat) that deviate from the overall trends. The recent Blowes et al. paper in Science covers many taxa and regions across the globe with the same conclusion. European data sets (the focus of the present study) are particularly well represented in previous analyses. None of this means that the study under review is not a meaningful contribution, but these considerations might be relevant to the editor in deciding whether Nature Communications is the right venue.

Response #21: Indeed Blowes et al. 2019 used an impressive global dataset of 51,392 biodiversity time series from the terrestrial, freshwater and marine realms, representing Europe particularly well. However there are significant differences in the data used: The mean length of the time series used in Blowes et al. is 5.5 years (median 6 years compared to 21 years in our study), while our shortest time series is 15 years (compared to 3 years in Blowes et al.). Furthermore, $\approx 44\%$ of the time series in Blowes et al. comprise just two sampling years and another $\approx 20\%$ just 3. This means approximately 2/3 of all times series in Blowes et al. were based on two or three data points. In contrast, we use datasets with at least 10 sampling years (only exceptions were a few vegetation datasets). Given the well-documented natural interannual variability in common biodiversity metrics, we believe that this is an important difference as a robust calculation of long-term trends requires much more than 2-3 data points. Furthermore, Blowes et al. included all data that have already been investigated in Dornelas et al. while none of our datasets have been included neither in Blowes et al. nor Dornelas et al. In addition Blowes et al. aggregated studies with a single location, and those with extents $< 71.7 \text{ km}^2$ within the respective grid cell (96 km^2) which affected $\approx 16\%$ of their time series while we did not aggregate data with different coordinates.

Blowes et al. calculated common community metrics, all based on presence/absence data. In our study, we additionally calculated abundance trends that are missing in Blowes et al. 2019 but are regarded to be important as changes in abundance are often an early indicator for more substantial changes within a community. Furthermore, for the first time in such a study, we used independent variables (naturalness, temperature, precipitation, etc.) in a statistical model to unravel potential drivers of the observed changes in biodiversity metrics, while Blowes et al. did

not use any environmental explanatory variables. Moreover, in a large paragraph of our Discussion we highlight the implications of our results for monitoring and managing biodiversity, another issue not covered by Blowes et al.

For all these reasons, we believe that our study is not only using completely independent data (so far unexplored and not included in BioTIME) with a much larger temporal extent (minimum 15 years, minimum 10 data points) and additional community metrics but also uncovers some novel aspects, for example: (a) no overall abundance trends in Europe (not investigated in neither Dornelas et al. nor in Blowes et al.), (b) the clear spatial structure of temporal turnover in Europe or (c) the correlations between abundance and richness trends with independent environmental explanatory variables such as temperature and naturalness. Furthermore our strict criteria regarding the time series included in our study have been confirmed by a recent study of Didham et al. 2020 where the authors are basically saying the same: more than 15 years and at least 10 data points.

(2) In terms of data limitations, the data here are of essentially the same nature as in BioTime (Blowes et al. and Dornelas et al.), and so they do not overcome previously noted limitations (doing so would more clearly represent an advance). While there are many regions, habitats, and taxa represented, the representation of combinations is actually fairly limited. Table S2 shows that roughly 17% of bioregion-realm-taxon combinations are represented, and less than 10% are represented by more than 1 study. This manifests clearly in the Discussion, where differences among regions or taxa are often interpreted with reference to a different variable (e.g., a pattern in a particular region is likely because that region only has data sets for particular taxa). Perhaps the most glaring example (line 209) is algae being represented by one river catchment, which probably means there should be no category “algae” in this analysis. So, while we can see that there is heterogeneity among time series, there is limited scope for drawing strong conclusions about trends for particular taxa or regions (i.e., what factors underlie the heterogeneity). As such, when the authors say (line 123) that “a comprehensive overview directly comparing trends among complementary datasets has been lacking so far”, it’s not so clear that this “lack” has been filled by the current study.

Response #22: As mentioned in our previous response, we believe that there are substantial differences between our data and the data used in Blowes et al. 2019. The nature of our data is closer to the data used in Dornelas et al. 2014, who used 100 long-term datasets across the globe with a median length of 14 years and 13 data points, but almost 1/3 of their time series were shorter than 10 years. Dornelas et al., however, did not use any data from continental Europe, nor did they incorporate independent environmental explanatory variables. This means that our study fills an important gap as it is the first to use truly long-term biodiversity data from Europe. In this respect our study “overcomes previously noted limitations” as we explicitly excluded shorter time series <15 years which represent the majority of datasets in both Dornelas et al. and Blowes et al.

We agree that our “representation of combinations is actually fairly limited” (“17% of bioregion-realm-taxon combinations are represented”). Ideally we would have extended the study to a more homogeneous representation of biogeoregions and biota, but unfortunately we could not

with the available datasets. We clearly present the distribution of studied datasets in the main text (see Figure 1) and in table S2. Moreover, we ran a sensitivity analysis to assess the effect of the unbalanced design on the results of the meta-analysis mixed models (Appendix A).

(3) In revising the manuscript, I think clarity could be improved with greater precision of several statements or arguments. For example:

(a) Community stability. This concept is emphasized in the abstract, but in the Methods it is revealed that this just means species turnover (specifically the inverse). So, a community in which composition is changing is considered “unstable”. Furthermore, species turnover is interpreted as “an early warning of more severe biodiversity changes to come”. I can’t think of any basis in the data for this. When environmental conditions change, fitness declines for some species and increases for others, such that turnover may well be a sign of adaptation (in the broad sense). I think a less value-laden presentation would simply call it turnover. Presenting possible interpretations in the Discussion is reasonable, but I see no basis for assuming there is any indication of “instability” or a “warning” of bad things to come.

Response #23: We have removed the term community stability from the manuscript. We have also rephrased the last two sentences of the Abstract and the respective sentences in the second paragraph of the discussion to make our manuscript less value-laden.

(b) Line 259: This line is hard to follow. “extended on a larger geographic area”. How so?

Response #24: We rephrased that sentence as: “Accordingly, we show that temporal changes in taxon turnover are more pertinent across biogeoregions than the other three studied biodiversity metrics.” (lines 263-265)

(c) Line 263: “turnover driven changes in community stability”. This doesn’t really make sense, since stability is defined as low turnover. One therefore can’t “drive” the other (one is the other). The subsequent list of explanations is also difficult to follow: e.g., “stochastic responses to climate change” (stochastic in what sense?) On line 266, it is not clear how compensatory dynamics is a different explanation from the others (one could be embedded in the other).

Response #25: As mentioned before, we removed any reference to “community stability”, and rephrased this part of our discussion (Lines 266-271).

(d) Line 318: “a detrimental effect in restoring actual trends in biodiversity”. Not clear what is meant by this.

Response #26: Rephrased as “can have a detrimental effect in restoration ecology” (lines 326-327).

(e) Line 253: What is meant by the word “masked”? One set of species declining and another increasing are two different things. If one looks specifically for either, they can find it. If they look at something else (e.g., net change in the number of species), they will not. But nothing is being “masked”.

Response #27: We have changed the sentence to “Furthermore, the loss of specialist taxa could be compensated locally by...” (line 257)

(f) There is ambiguity with respect to whether temperature is treated as an average or a trend (it says different things in different places).

Response #28: We used the S-statistic of temperature trends in the meta-analysis mixed models. We have clarified it in the text of the Result section (paragraph 3) and in Table 2.

Reviewers' Comments:

Reviewer #1:

Remarks to the Author:

The paper "Changes in long-term biodiversity trends in Europe" is an important contribution to the ongoing debate about biodiversity loss. It presents and synthesizes new data across a broad range of taxa and its differentiated analyses of region, taxa etc is novel and of broad interest to readers. I suggest accepting it after addressing only a few more comments listed below and I suggest citing the new study by van Klink et al which appeared last week.

I 106: maybe specify: terrestrial insects

I 135: remove brackets

I 234: please use similar terminology as in line 230

I 269: I don't fully understand why these two sentences are contrasted so strongly. I think instead of 'However, there is an alternative explanation.' a simple 'Moreover,...' is sufficient.

I 275: It is true pollution and other stressors have weakened in Eastern Europe. On the other hand, Eastern Europe had maintained more habitat of high naturalness (e.g. pristine forests, small scale agriculture of low land-use intensity) which has (partly) experienced serious changes with increasing land-use intensification (see e.g. Kümmerle et al 2016 *Env Res Letters: Agricultural intensification in the 19th and 20th century also began later and progressed slower in Europe's East than in its West* (Jepsen et al 2015). thus, yields gaps are higher in Eastern Europe and intensification easier, likely in part explaining the patterns of stronger agriculture intensification we found for some Eastern Europe regions (figure 4).) I suggest to phrase lines 266-278 a bit more widely, providing both your potential explanations plus land-use changes as third one as equally possible or even interacting ones.

I 294: L. 291-297: It is important to point out that results from one group can not be extrapolated to others. But I think for a balanced presentation of results and examples, it is very important to add here that terrestrial invertebrates decreased in abundance in the Atlantic region - corroborating earlier results. Media and public attention to this topic is high making a balanced presentation extremely important.

I 320: do you really mean 1990?

I 337: trends

Reviewer #2:

Remarks to the Author:

I appreciate the authors' effort on revising the manuscript and feel that the authors have managed well in general to address the points I raised. I believe that the clarity of the manuscript has now been improved which the readers will appreciate greatly.

REVIEWERS' COMMENTS:

Reviewer #1 (Remarks to the Author):

The paper "Changes in long-term biodiversity trends in Europe" is an important contribution to the ongoing debate about biodiversity loss. It presents and synthesizes new data across a broad range of taxa and its differentiated analyses of region, taxa etc is novel and of broad interest to readers. I suggest accepting it after addressing only a few more comments listed below and I suggest citing the new study by van Klink et al which appeared last week.

Authors' response: Firstly, we would like to thank the reviewer for their positive comments regarding our manuscript! We have included all the requested changes; however, we are slightly reluctant to cite the new study by van Klink et al. While we clearly see a link between their study and ours, many scientists have already issued criticisms regarding the suitability of several datasets used in their analyses and the general conclusions drawn in the discussion. Moreover, we have already reached the maximum number of allowed reference (70). Nevertheless, should the editor require, we will cite this paper in our manuscript.

I 106: maybe specify: terrestrial insects

Authors' response: Done.

I 135: remove brackets

Authors' response: Done.

I 234: please use similar terminology as in line 230

Authors' response: Done.

I 269: I don't fully understand why these two sentences are contrasted so strongly. I think Instead of 'However, there is an alternative explanation.' a simple 'Moreover,...' is sufficient.

Authors' response: We changed that part as suggested by the reviewer.

I 275: It is true pollution and other stressors have weakened in Eastern Europe. On the other hand, Eastern Europe had maintained more habitat of high naturalness (e.g. pristine forests, small scale agriculture of low land-use intensity) which has (partly) experienced serious changes with increasing land-use intensification (see e.g. Kümmerle et al 2016 *Env Res Letters*: Agricultural intensification in the 19th and 20th century also began later and progressed slower in Europe's East than in its West (Jepsen et al 2015). thus, yields gaps are higher in Eastern Europe and intensification easier, likely in part explaining the patterns of stronger agriculture intensification we found for some Eastern Europe regions (figure 4).) I suggest to phrase lines 266-278 a bit more widely, providing both your potential explanations plus land-use changes as third one as equally possible or even interacting ones.

Authors' response: Following the suggestion of the reviewer, we added this sentence: "Another possible interacting factor is the change in land use, which follows different temporal trajectories in different European regions, and thus could concur in explaining regional differences in biodiversity trends"

I 294: L. 291-297: It is important to point out that results from one group can not be extrapolated to others. But I think for a balanced presentation of results and examples, it is very important to add here that terrestrial invertebrates decreased in abundance in the Atlantic region - corroborating earlier results. Media and public attention to this topic is high making a balanced presentation extremely important.

Authors' response: We followed the suggestion of the reviewer, and added this sentence: "On the other hand, we recorded declines in species abundances for terrestrial invertebrates in the Atlantic biogeoregion, consistent with previous findings^{31,32,50}".

I 320: do you really mean 1990?

Authors' response: We have corrected it to "1980s".

I 337: trends

Authors' response: We have corrected it.

Reviewer #2 (Remarks to the Author):

I appreciate the authors' effort on revising the manuscript and feel that the authors have managed well in general to address the points I raised. I believe that the clarity of the manuscript has now been improved which the readers will appreciate greatly.

Authors' response: Thank you.